# Human single-neuron activity is modulated by intracranial theta burst stimulation of the basolateral amygdala

Justin M Campbell[1]*, Rhiannon L Cowan[2], Krista L Wahlstrom[3], Martina K Hollearn[3], Dylan Jensen[1], Tyler Davis[2], Shervin Rahimpour[2], Ben Shofty[2], Amir Arain[4], John D Rolston[5], Stephan Hamann[6], Shuo Wang[7], Lawrence N Eisenman[8], James Swift[9,10], Tao Xie[9,10], Peter Brunner[9,10], Joseph Manns[6], Cory Inman[1,3†], Elliot H Smith[2†], Jon Timothy Willie[9,10†]

[1]Interdepartmental Program in Neuroscience, University of Utah, Salt Lake City, United States; [2]Department of Neurosurgery, University of Utah, Salt Lake City, United States; [3]Department of Psychology, University of Utah, Salt Lake City, United States; [4]Department of Neurology, University of Utah, Salt Lake City, United States; [5]Department of Neurosurgery, Brigham and Women's Hospital, Boston, United States; [6]Department of Psychology, Emory University, Atlanta, United States; [7]Department of Radiology, Washington University School of Medicine, St. Louis, United States; [8]Department of Neurology, Washington University School of Medicine, St. Louis, United States; [9]Department of Neurological Surgery, Washington University School of Medicine, St. Louis, United States; [10]National Center for Adaptive Neurotechnologies, St. Louis, United States

*For correspondence: justin.campbell@hsc.utah.edu

†These authors contributed equally to this work

## eLife Assessment

This **important** study provides a description of how single-neuron firing rates in the human medial temporal lobe and frontal cortex are modulated by theta-burst stimulation of the basolateral amydala. The results are supported by **convincing** evidence obtained from a rigorous task design and analysis of an incredibly rare dataset. The results may help guide future studies incorporating amygdala stimulation to improve patient health.

**Abstract** Direct electrical stimulation of the human brain has been used for numerous clinical and scientific applications. At present, however, little is known about how intracranial stimulation affects activity at the microscale. In this study, we recorded intracranial EEG data from a cohort of patients with medically refractory epilepsy as they completed a visual recognition memory task. During the memory task, brief trains of intracranial theta burst stimulation (TBS) were delivered to the basolateral amygdala (BLA). Using simultaneous microelectrode recordings, we isolated neurons in the hippocampus, amygdala, orbitofrontal cortex, and anterior cingulate cortex and tested whether stimulation enhanced or suppressed firing rates. Additionally, we characterized the properties of modulated neurons, clustered presumed excitatory and inhibitory neurons by waveform morphology, and examined the extent to which modulation affected memory task performance. We observed a subset of neurons (~30%) whose firing rate was modulated by TBS, exhibiting highly heterogeneous responses with respect to onset latency, duration, and direction of effect. Notably, location and baseline activity predicted which neurons were most susceptible to modulation, although the impact of this neuronal modulation on memory remains unclear. These findings advance our limited understanding of how focal electrical fields influence neuronal firing at the single-cell level.

## Introduction

The amygdala is a highly connected cluster of nuclei with input from multiple sensory modalities and vast projections to distributed cortical and subcortical regions involved in autonomic regulation and cognition (*Kim et al., 2011*; *McDonald and Mott, 2017*; *Roy et al., 2009*; *Stein et al., 2007*). Numerous studies have described the amygdala's capacity to facilitate the encoding of long-lasting emotional memories (*Dolcos et al., 2004*; *Hermans et al., 2014*; *LaBar and Cabeza, 2006*; *Phelps, 2006*; *Phelps, 2004*; *Phelps and LeDoux, 2005*; *Qasim et al., 2023*; *Richardson et al., 2004*; *Roozendaal et al., 2009*; *Schwabe et al., 2013*; *Zheng et al., 2017*). Specific oscillatory dynamics in the amygdalohippocampal circuit are increasingly understood to be essential in prioritizing the encoding of these salient memories (*Qasim et al., 2023*; *Zheng et al., 2019*; *Zheng et al., 2017*).

Recently, direct electrical stimulation of the basolateral complex of the amygdala (BLA) in humans has revealed a more generalized ability to enhance declarative memory irrespective of the emotional valence (*Inman et al., 2018*), likely by promoting synaptic plasticity-related processes underlying memory consolidation in the hippocampus and medial temporal lobe (*McGaugh, 2013*; *McGaugh, 2004*; *McGaugh et al., 2002*; *Roesler et al., 2021*). These pro-memory effects were achieved with rhythmic theta burst stimulation (TBS), which is known to induce long-term potentiation (LTP), a key mechanism in memory formation (*Larson and Munkácsy, 2015*). Emerging evidence suggests that intracranial TBS may also enhance memory specificity (*Titiz et al., 2017*), evoke theta-frequency oscillations (*Solomon et al., 2021*), and facilitate short-term plasticity in local field potential recordings (*Herrero et al., 2021*; *Huang et al., 2024*). However, the extent to which exogenous TBS modulates activity at the single-cell level and whether this modulation is associated with memory performance remains poorly understood.

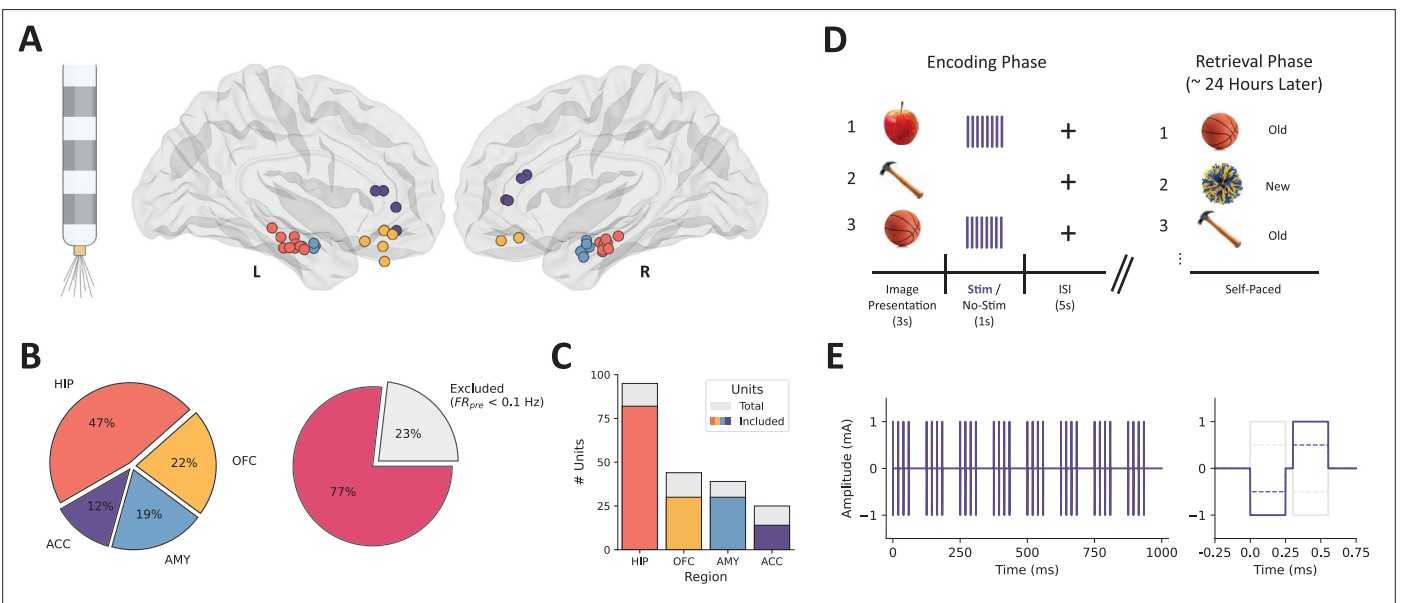

**Figure 1.** Microelectrode locations, unit counts, and experimental design. (**A**) Behnke Fried-style macro/micro depth electrode (left) and microelectrode bundle locations projected in MNI space (right). (**B**) The proportion of units recorded from each brain area (left) and the proportion of units that met the criteria for inclusion in analyses (average pre-trial baseline firing rate ≥0.1 Hz) (right). (**C**) Counts of total (gray) and included (colored) units within each region. (**D**) Intracranial recording and stimulation took place in the context of a two-phase (encoding, retrieval) visual recognition memory task. A series of neutral valence images were shown (3 s), half of which were followed by direct electrical stimulation (1 s). Retrieval memory was tested during a self-paced task ~24 hr later. (**E**) Simulated theta burst stimulation trace (left) and individual stimulation pulse (right); charge-balanced, bipolar, biphasic rectangular pulses were delivered over a 1 s period. HIP = hippocampus (coral), OFC = orbitofrontal cortex (yellow), AMY = amygdala (blue), ACC = anterior cingulate cortex (purple).

The online version of this article includes the following figure supplement(s) for figure 1:

**Figure supplement 1.** Anatomical location of stimulated electrodes.

**Figure supplement 2.** Unit quality metrics.

**Figure supplement 3.** Characterization of units based on laterality relative to stimulation.

Here, we address this knowledge gap by conducting simultaneous microelectrode recordings from prefrontal and medial temporal structures during a memory task in which intracranial TBS was applied to the BLA. To characterize neuronal modulation, we contrasted the trial-averaged, peri-stimulation firing rates across stim and no-stim conditions; neurons were further classified based on their location, baseline activity, and direction of effect (enhancement vs. suppression). We predicted that intracranial TBS of the BLA would modulate spiking activity within highly connected regions (e.g. hippocampus) and improve memory task performance.

## Results

We recorded single-unit activity from 23 patients (n=30 sessions) with medically refractory epilepsy as they completed a visual recognition memory task (see *Supplementary file 1* for patient demographics). During the encoding session of the experiment, each patient received either 80 or 160 trials of bipolar intracranial TBS to a contiguous pair of macroelectrode contacts in the BLA (see *Figure 1—figure supplement 1* for anatomical localization of stimulated contacts). An equal number of 'no-stimulation' trials were randomly interspersed to evaluate the effect of stimulation on memory performance and control for neuronal modulation resulting from experimental stimuli (e.g. image presentation). In total, we isolated 203 putative neurons from 68 bundles of 8 microwires each, distributed among recording sites in the hippocampus (HIP, n=95 units), orbitofrontal cortex (OFC, n=44), amygdala (AMY, n=39), and anterior cingulate cortex (ACC, n=25) (*Figure 1*; see also *Figure 1—figure supplement 2* and *Figure 1—figure supplement 3*); a subset of these units (n=47, 23.2%) was excluded from subsequent analyses because low baseline firing rates (<0.1 Hz) limited the ability to robustly detect modulation.

### TBS of the BLA modulates widely distributed populations of neurons

We hypothesized that BLA stimulation would modulate neuronal activity in the sampled regions, given the amygdala's well-established connectivity to the HIP, OFC, and ACC (*Roy et al., 2009*). To test this hypothesis, we quantified spike counts across trials within peri-stimulation epochs (1 s pre-trial ISI, 1 s after image onset, 1 s during stimulation/after image offset, and 1 s post-stimulation) and used Wilcoxon signed-rank tests to compare the spike counts against a null distribution generated by shuffling epoch labels. We performed two firing rate contrasts across trials (pre-trial ISI vs. during stimulation, pre-trial ISI vs. post-stimulation) within two distinct conditions (stim, no-stim); an additional contrast of the pre-trial ISI vs. image onset epochs was included to evaluate the sensitivity of neurons to task image presentations (*Figure 2A*).

BLA TBS modulated firing rates in 30.1% of all recorded units, a significantly higher proportion than the 15.4% responsive to no-stim (image only) trials (one-sided Fisher's exact test, OR = 2.37, p<0.001; *Figure 3A*). Across all regions sampled, we observed units modulated by the stim and no-stim conditions. Units in HIP (OR = 2.07, p=0.044), OFC (OR = 5.09, p=0.040), and AMY (OR = 3.33, p=0.042) were most sensitive to stimulation; we did not observe a difference in the proportions of units within the ACC responsive to the stim vs. no-stim conditions (OR = 1.00, p=0.661; *Figure 3B*). Only 9.0% of units responded to both the stim and no-stim conditions, despite approximately half of the stim-modulated units (representing 14.7% of all units) exhibiting a change in firing rate associated with image onset (*Figure 3D*). This result suggests that the units modulated by stimulation are largely distinct from those responsive to image offset during trials without stimulation. Stimulation, however, did not appear to alter the rhythmicity in neuronal firing, as measured by spiking autocorrelograms (*Figure 3—figure supplement 1*).

Because neuronal firing properties vary across cell types (*Barthó et al., 2004*; *Keller et al., 2010*; *Peyrache et al., 2012*; *Le Van Quyen et al., 2008*), we also tested whether baseline (pre-trial ISI) firing rates predicted a unit's response to stimulation, suggestive of selective engagement of specific neuron populations. Stimulation-modulated units exhibited significantly higher baseline firing rates compared to unaffected units ($U$(N$_{Stim, Mod}$ = 47, N$_{Stim, NS}$ = 109)=3450.50, p<0.001). No difference in baseline firing rate was observed among units modulated in the no-stim condition, compared to those that were unaffected ($U$(N$_{No-Stim, Mod}$ = 24, N$_{No-Stim, NS}$ = 132)=1964.00, p=0.062) (*Figure 3C*). The median (Q1, Q3) baseline firing rates for modulated units in the stim and no-stim conditions were 1.77 Hz (0.95 Hz, 5.39 Hz) and 1.53 Hz (0.72 Hz, 5.41 Hz), respectively.

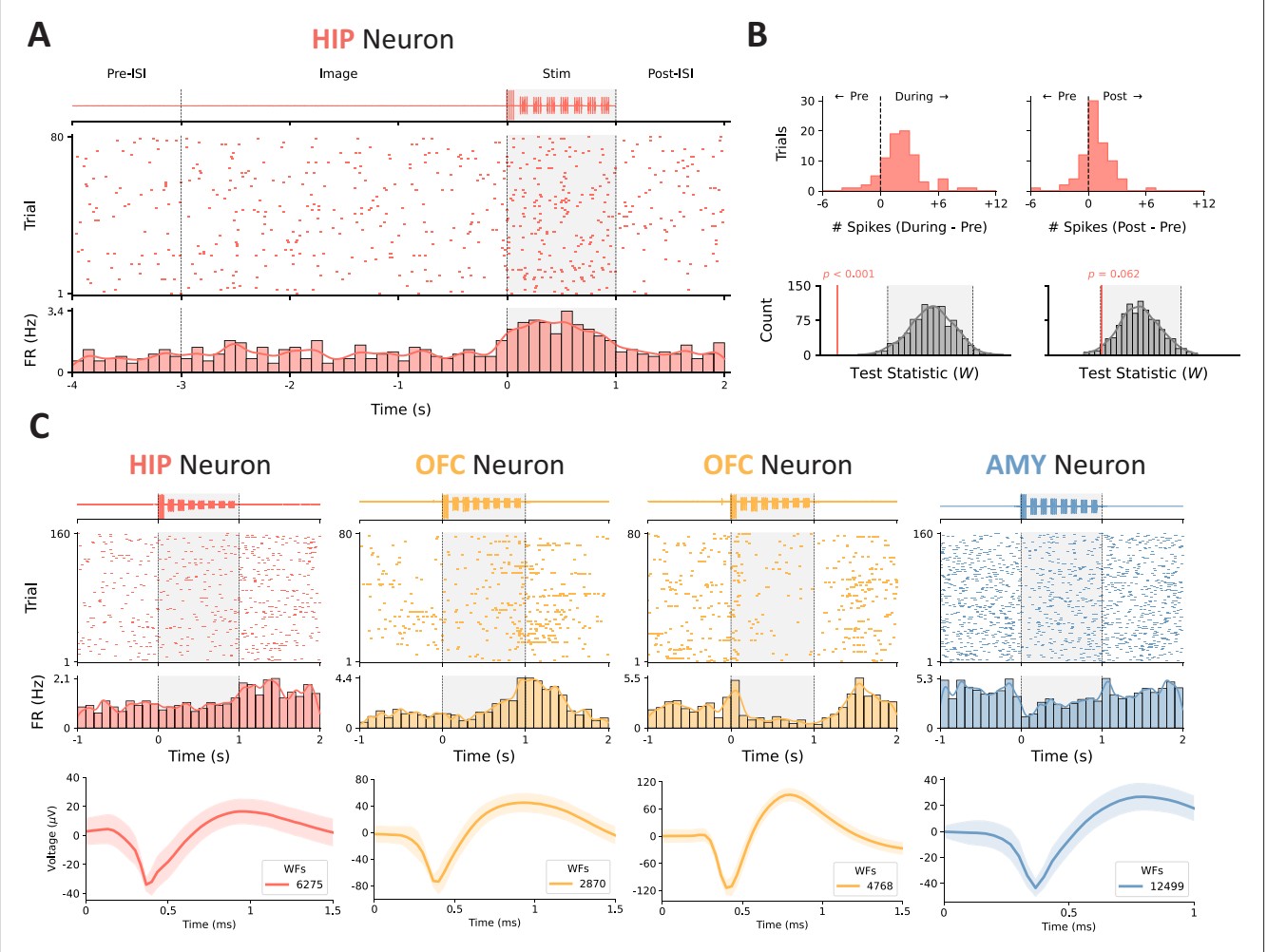

**Figure 2.** Example raster plots depicting heterogeneous responses to stimulation. (**A**) Representative example of modulation during stimulation. The high-pass-filtered, trial-averaged LFP from the corresponding microwire is shown (top) above the spike raster for an example unit located in the hippocampus (middle); the gray shaded region depicts the duration of stimulation with onset at t=0. The average firing rate across trials was estimated by convolving the binned spike counts (100 ms bins) with a Gaussian kernel (bottom). (**B**) The difference in the number of spikes in the 1 s peri-stimulation epochs for each trial is shown (top). We subsequently performed a Wilcoxon signed-rank test on the during- and post-stimulation spike counts for each trial vs. the pre-trial baseline and compared the empirical test statistic against a null distribution generated by shuffling the epoch labels 1000 times (bottom); the gray-shaded region represents the distribution containing 95% of observed values. (**C**) Some units (left, left-middle) exhibited increased firing rates, whereas others (right-middle, right) had their firing suppressed. The temporal dynamics of the firing rate modulation (e.g. onset, duration) were highly variable across units. The averaged waveform for each of the visualized units is shown below its corresponding peri-stimulation raster plot (WFs = waveforms); the shaded region represents standard deviation across waveforms.

The online version of this article includes the following figure supplement(s) for figure 2:

**Figure supplement 1.** Control analyses for the detection of modulated units.

In a subset of experimental sessions (n=7), we explored the effects of different stimulation parameters on neuronal modulation within an experimental session; more specifically, we employed a lower stimulation amplitude (0.5 mA vs. 1.0 mA) and varied pulse frequency (33 Hz vs. 50 Hz and 33 Hz vs. 80 Hz). Neither the amplitude of stimulation (OR = 1.69, p=0.302, n=30) nor pulse frequency (33 vs. 80 Hz; OR = 0.00, p=1.000, n=1; 50 vs. 80 Hz; OR = 1.40, p=0.758, n=6) significantly altered the proportion of modulated units (*Figure 3—figure supplement 2*).

Our exploratory analyses of pseudo-population activity revealed interesting temporal dynamics associated with image presentation and the delivery of stimulation. More specifically, we observed variation among the first three principal components across both stim and no-stim trials associated with image presentation (t = –3 to t=0) and robust shifts in coactivity modes associated with

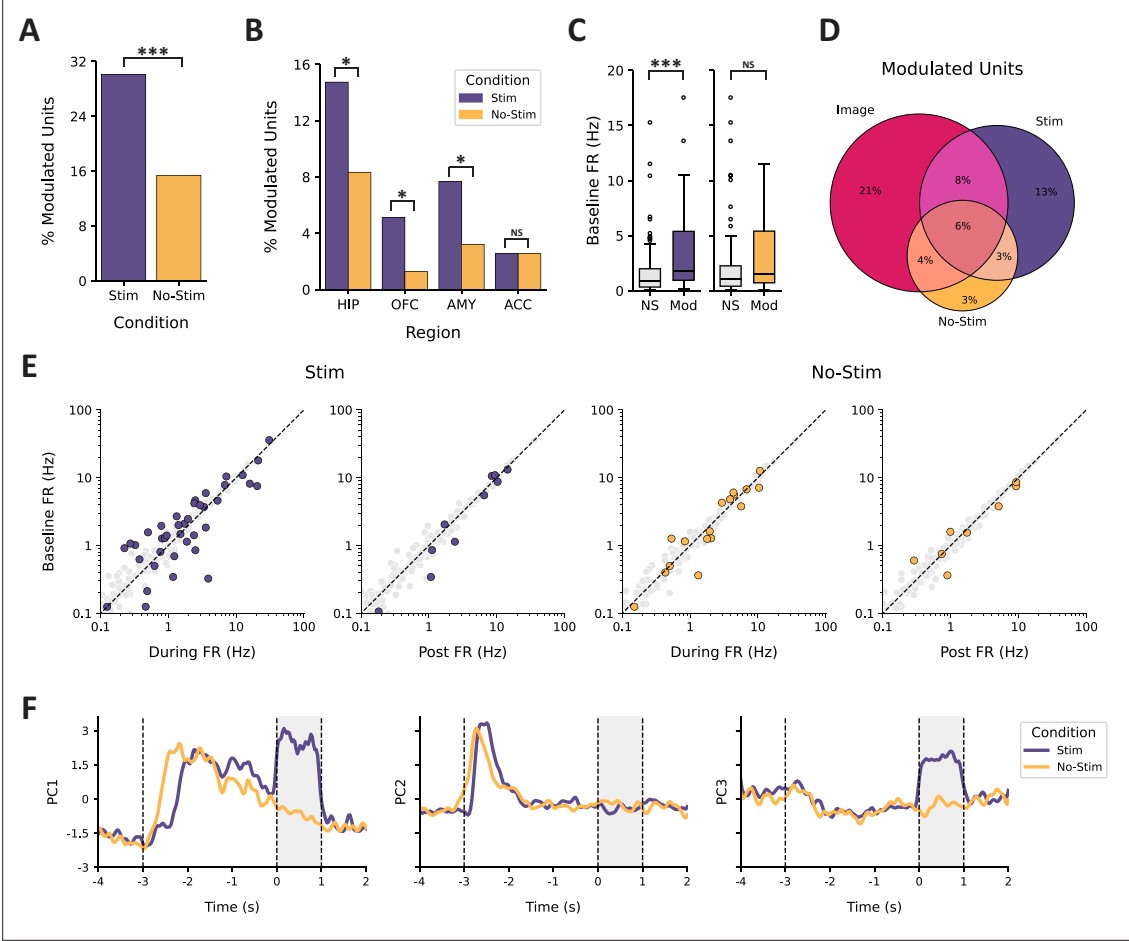

**Figure 3.** Characterization of modulation in neuronal firing rate. (**A**) Percent of modulated units observed across trials separated by stim (purple) vs. no-stim conditions (orange). (**B**) Percent of modulated units as a function of recording region. (**C**) Comparison of baseline firing rate in units separated by condition (stim vs. no-stim) and outcome (NS = not significant, Mod = modulated). (**D**) Venn diagram depicting the shared and independent proportions of units modulated by image onset (Image) and the two experimental conditions (stim vs. no-stim). (**E**) Scatterplot of pre-stimulation firing rate relative to the firing rate during the two contrast windows (during, post) for the stim (left) and no-stim (right) conditions. Modulated units are highlighted in purple (stim) or orange (no-stim), whereas units without a significant change are shown in gray. (**F**) Temporal dynamics of pseudo-population coactivity within each condition, represented by the first three principal components of the trial-averaged firing rates. The gray-shaded region depicts the duration of stimulation with onset at t=0. Images were presented on screen for 3 s, with onset at t = –3. * p<0.05, *** p<0.001, NS = not significant.

The online version of this article includes the following figure supplement(s) for figure 3:

**Figure supplement 1.** Analysis of modulation in spiking rhythmicity.

**Figure supplement 2.** Sub-analysis of stimulation parameters used across experiments.

**Figure supplement 3.** Analysis of pseudo-population activity within regions, separated by laterality relative to stimulation.

**Figure supplement 4.** Analysis of multiunit activity (MUA) response to stimulation.

stimulation (t=0 to t=1) (*Figure 3F*). Further characterization of these dynamics suggests that TBS of the BLA primarily resulted in transient changes to firing rate coactivity within hippocampal neurons and was present regardless of the neuron's laterality to stimulation (*Figure 3—figure supplement 3*). Taken together, these analyses reveal global structure in the state space of responses to BLA stimulation within hippocampal circuits.

Finally, we performed three supplementary analyses to evaluate the robustness of our approach to detecting firing rate modulation: a sensitivity analysis assessing the proportion of modulated units at different firing rate thresholds for inclusion/exclusion, a data dropout analysis designed to control for the possibility that nonphysiological stimulation artifacts may preclude the detection of temporally adjacent spiking, and a synthetic detection probability analysis. These results recapitulate our observation that units with higher baseline firing are most likely to exhibit modulation (though the

probability of detecting modulation is lower for sparsely active neurons) and suggest that suppression in firing rate is not solely attributable to amplifier saturation following stimulation (*Figure 2—figure supplement 1*).

## Neurons exhibit heterogeneous responses to TBS

Recent studies have reported enhanced neural plasticity (via intracranial local field potential recordings and evoked responses) following repetitive direct electrical stimulation (*Herrero et al., 2021*; *Huang et al., 2024*; *Huang et al., 2019*; *Keller et al., 2018*). Accordingly, we hypothesized that recorded units would predominantly exhibit enhanced spiking in response to intracranial TBS of the BLA. Similarly, individual units exhibited highly variable responses to stimulation with respect to onset latency (rapid vs. delayed), duration (transient vs. durable), and valence (enhancement vs. suppression) (*Figure 2B*).

The most common epoch for firing rate modulation was during the 1 s epoch in which TBS was delivered (25.0% of all neurons). Smaller subsets were modulated only in the 1 s post-stimulation epoch (6.4%) or in both the during- and post-stimulation epochs (1.3%). A similar trend was observed for modulation in the no-stim condition: 10.9% during, 5.8% post, and 1.3% for both. Suppression was most common among modulated units during stimulation (56.4%), whereas enhancement was the dominant response post-stimulation (70.0%). In contrast, enhancement was most common within both epochs across no-stim trials (58.5% during, 66.7% post). The mean (± SD) absolute z-scored difference in firing rate across stimulation trials (relative to pre-trial ISI) was z=0.60 (±0.58) and z=0.43

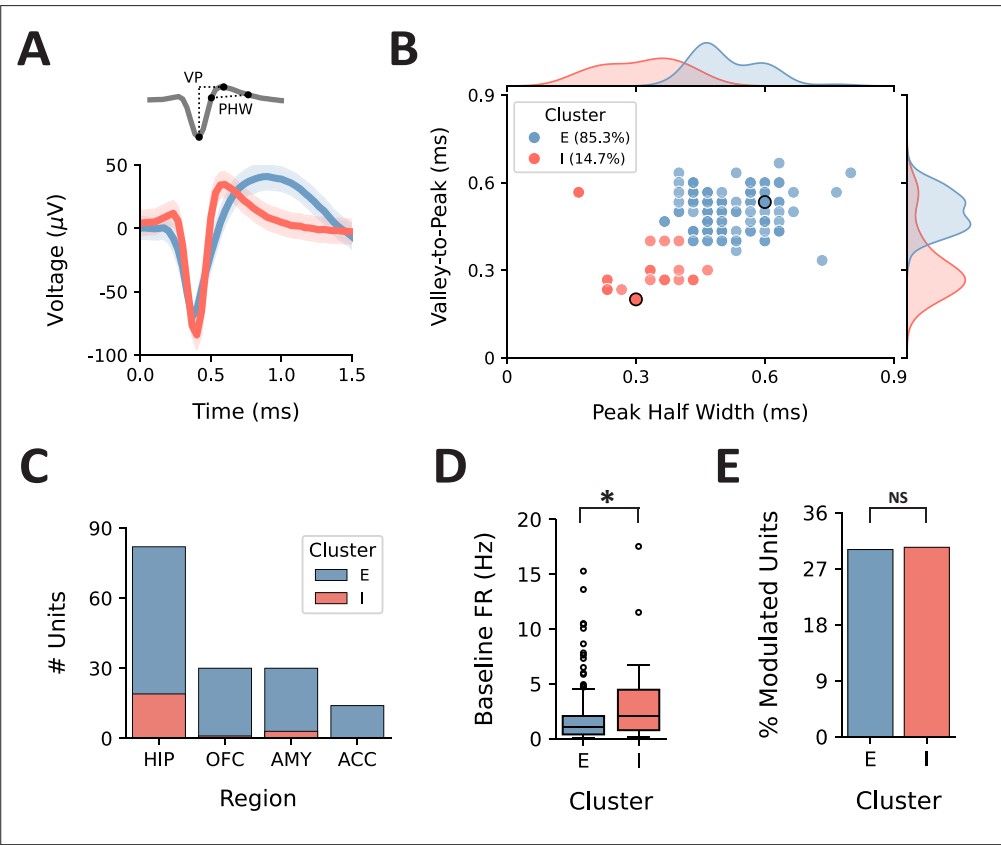

**Figure 4.** Separation of presumed excitatory and inhibitory neurons by waveform morphology. (**A**) Two metrics were calculated using the averaged waveforms for each detected unit: the valley-to-peak width (VP) and peak half-width (PHW). (**B**) Scatterplot of the relationship between VP and PHW; note that units with identical metrics are overlaid. Using k-means clustering, we identified two distinct response clusters, representing presumed excitatory (E, blue) and inhibitory (I, red) neurons. The units from which the example waveforms were taken are outlined in black. Probability distributions for each metric are shown along the axes. (**C**) Total number of units within each cluster, separated by region. (**D**) Comparison of baseline firing rates, separated by cluster. (**E**) Percent of modulated units in each cluster. *p<0.05, NS = not significant.

(±0.27) for the during- and post-stimulation epochs, respectively. Across no-stim trials, we observed a mean absolute z-scored difference of z=0.38 (±0.24) and z=0.30 (±0.18) in analogous epochs (*Figure 3E*). Additional characterization of multiunit activity (MUA) revealed a dominant signature of increased activity post- vs. pre-stimulation, in line with the trends observed at the single-neuron level (*Figure 3—figure supplement 4*).

## TBS modulates excitatory and inhibitory neurons equally

Using k-means clustering, we grouped neurons into two distinct clusters based on waveform morphology, representing neurons that were presumed to be excitatory (E) and inhibitory (I) (*Figure 4B*). Inhibitory (fast-spiking) neurons exhibited shorter waveform valley-to-peak width (VP) and peak half-width (PHW), compared with excitatory (regular-spiking) neurons (I cluster centroid: VP = 0.50 ms, PHW = 0.51 ms; E cluster centroid: VP = 0.32 ms, PHW = 0.31 ms) and greater baseline firing rates ($U$($N_I$ = 23, $N_E$ = 133)=1074.50, p=0.023) (*Figure 4D*). Although we observed a much greater proportion of excitatory vs. inhibitory neurons (E: 85.3%, I: 14.7%), stimulation appeared to affect excitatory and inhibitory neurons equally, suggesting that one cell type is not preferentially activated over another (*Figure 4E*).

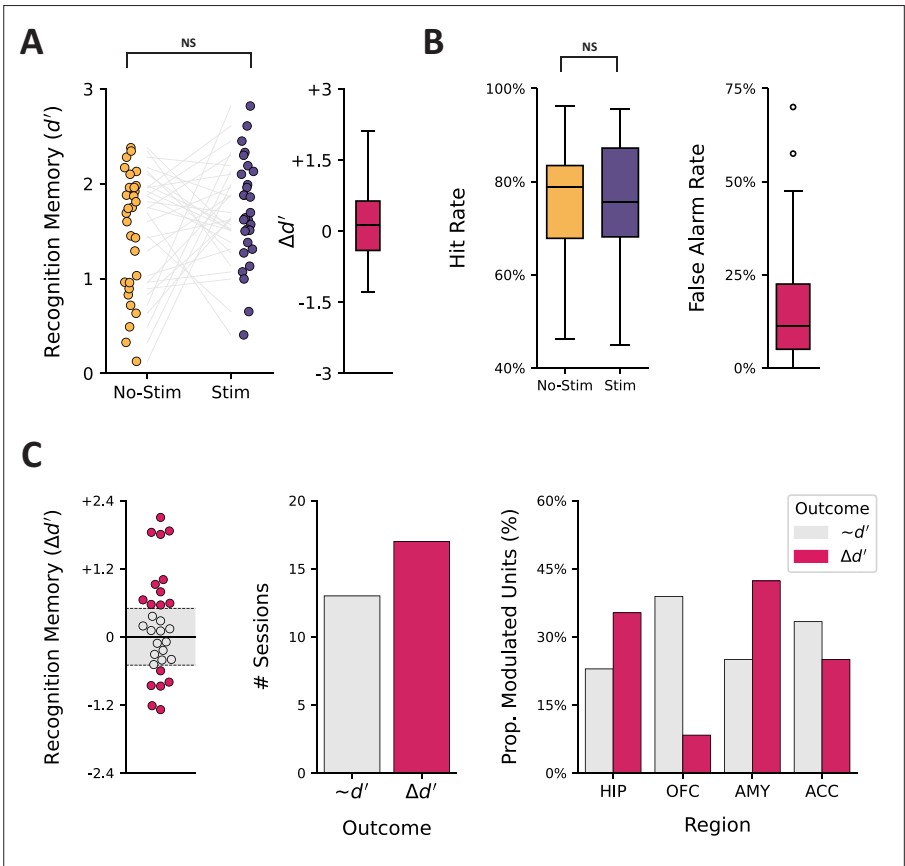

**Figure 5.** Summary of behavioral performance during memory task. (**A**) Memory performance for each session is quantified using d' (left); gray lines connect d' scores across conditions for an individual session. Boxplot of the observed difference in d' scores across conditions (right). (**B**) Hit rate (percent of old images correctly recognized) and false alarm rate (percent of new images incorrectly labeled as old) across conditions. (**C**) Change in recognition memory performance was split into two categories using a d' difference threshold of ±0.5: responder (positive or negative; Δd', pink) and non-responder (~d', gray). Individual d' scores are shown (left) with points colored by outcome category; dotted lines demarcate category boundaries, and the gray-shaded region represents negligible change. The number of sessions within each outcome category (middle) and the proportion of modulated units as a function of outcome category, separated by region (right). NS = not significant.

The online version of this article includes the following figure supplement(s) for figure 5:

**Figure supplement 1.** The effect of stimulation proximity to white matter and distance to recorded neurons.

## Association between neuronal modulation and memory performance is unclear

Next, we investigated the link between stimulation and performance during the visual recognition memory task. To this end, we first used a linear mixed-effects model to examine the effect of condition (stim, no-stim) on memory performance (d') across trials in each session, with individual sessions treated as a random effect (intercept). Experiment type was also included as a fixed effect since data were aggregated across four highly similar experiments with minor differences in the content of visual stimuli, number of trials, stimulation parameters, and testing intervals. We did not observe an overall effect of memory enhancement (p>0.05) when controlling for subject-level variability (*Figure 5A*). The lack of a memory enhancement effect may be associated with high hit rates limiting sensitivity (mean ± SD) (75.7% ± 13.5% for no-stim trials, 75.0% ± 14.3% for stim trials) and considerable variability among false alarm rates across participants (17.9% ± 17.4%, range 0–70%; *Figure 5B*).

At the level of individual sessions, we observed enhanced memory (Δd' >+0.5) in 36.7%, impaired memory (Δd'<–0.5) in 20.0%, and negligible change (–0.5 ≤ Δd'≤0.5) in 43.3% when comparing performance between the stim and no-stim conditions; a threshold of Δd'±0.5 was chosen for this classification based on the defined range of a 'medium effect' for Cohen's d. To test our hypothesis that neuronal modulation would be associated with changes in memory performance, we combined the sessions that resulted in either memory enhancement or impairment and contrasted the proportion of modulated units across regions sampled. We did not, however, observe a meaningful difference in the proportion of modulated units when grouped by behavioral outcome (all contrasts p>0.05) (*Figure 5C*).

Proximity to white matter has been shown to influence the effects of stimulation on behavior and the strength of evoked responses (*Mankin et al., 2021*; *Mohan et al., 2020*; *Paulk et al., 2022*). Across all stimulated contacts, we observed only small differences in the proximity of stimulation contacts to white matter (median = 4.6 mm, range = 1.5–8.0 mm), likely because the chosen target (i.e. BLA) has several nearby white matter structures (e.g. stria terminalis). Nonetheless, we performed a linear regression between the proximity to white matter and the stimulation-induced effect on behavior (stimulation vs. no-stimulation d' difference), the results of which indicate no clear association (p>0.05; see *Figure 5—figure supplement 1*).

## Discussion

TBS is an efficient and validated paradigm for inducing LTP in neural circuits (*Larson and Munkácsy, 2015*). Additionally, intracranial TBS was recently shown to promote region-specific short-term plasticity (*Herrero et al., 2021*; *Huang et al., 2024*) and entrain frequency-matched oscillations (*Solomon et al., 2021*). At present, however, there is an incomplete understanding of how these population-level responses to stimulation relate to a modulation in the activity of individual neurons, which are thought to be the substrate of memory encoding and retrieval (*Hebb, 1949*). Here, we address this knowledge gap by characterizing neuronal firing recorded from microelectrodes in humans undergoing intracranial TBS of the BLA. Our experimental design focused explicitly on stimulation of the BLA, given our prior work that seeks to understand amygdala-mediated memory enhancement in humans (*Inman et al., 2023*; *Inman et al., 2020*; *Inman et al., 2018*).

We observed neurons distributed throughout the hippocampus, orbitofrontal cortex, anterior cingulate cortex, and amygdala responsive to direct electrical TBS. The effect of TBS on firing rate was heterogeneous with respect to onset, duration, and valence. Previous work characterizing local field potential responses to intracranial TBS observed similarly bidirectional effects throughout the brain suggestive of short-term plasticity (*Huang et al., 2024*). Few studies, however, have characterized the impact of exogenous stimulation on the spiking activity of individual neurons, none of which involve intracranial TBS. One study reported a long-lasting reduction in neural excitability among parietal neurons, with variable onset time and recovery following continuous transcranial TBS in nonhuman primates (*Romero et al., 2022*). In a similar vein, it was recently shown that human neurons are largely suppressed by single-pulse electrical stimulation (*Cowan et al., 2024*; *van der Plas et al., 2024*). Other emerging evidence suggests that transcranial direct current stimulation may entrain the rhythm rather than rate of neuronal spiking (*Krause et al., 2019*) and that stimulation-evoked modulation of spiking may meaningfully impact behavioral performance on cognitive tasks (*Fehring et al., 2024*).

An alternative approach has focused on the delivery of spatially selective microstimulation resembling the extracellular currents that normally modulate neuronal activity—this methodology has been used to bidirectionally drive neuronal firing in human temporal cortex (*Youssef et al., 2023*) and enhance memory specificity for images following stimulation (*Titiz et al., 2017*).

Although subsets of neurons from each region we sampled were responsive to stimulation, we observed the greatest difference in the proportion of modulated units across conditions in the hippo-campus, orbitofrontal cortex, and amygdala. This regional selectivity is to be expected, given that numerous studies have characterized how structural, functional, and effective connectivity among brain regions predicts the effects of stimulation (*Fox et al., 2020*; *Huang et al., 2024*; *Huang et al., 2019*; *Keller et al., 2018*; *Keller et al., 2011*; *Solomon et al., 2018*; *Stiso et al., 2019*). We also observed that units with greater baseline activity were most likely to exhibit modulated firing rates following stimulation. Other studies have identified firing patterns and waveform properties that differ between inhibitory and excitatory neurons in humans (*Barthó et al., 2004*; *Keller et al., 2010*; *Peyrache et al., 2012*; *Le Van Quyen et al., 2008*). For example, baseline firing rate disambigu-ates 'regular-spiking' and 'fast-spiking' neurons, which are presumed to represent pyramidal cells and interneurons, respectively. To test these hypotheses directly, we clustered neurons into presumed excitatory and inhibitory neurons based on waveform morphology. In doing so, we observed ~85% excitatory and ~15% inhibitory neurons, which is very similar to what has been reported previously in human intracranial recordings (*Cowan et al., 2024*; *Peyrache et al., 2012*). Interestingly, stimula-tion appeared to modulate approximately the same proportion of neurons for each cell type (~30%), despite the differently sized groups. Recent reports, however, have suggested that the extent to which electrical fields entrain neuronal spiking, particularly with respect to phase-locking, may be specific to distinct classes of cells (*Lee et al., 2024*).

Modulation in neuronal activity was defined by contrasting firing rates before, during, and after TBS across trials. In doing so, we were able to characterize coarse differences in activity indicative of enhancement or suppression. This approach, however, did not allow for analysis of more subtle, nuanced effects such as entrainment of spiking to individual bursts or pulses of TBS. Characterizations of rhythmicity in firing were challenging, given that most of the neurons we identified exhibit sparse activity with low baseline firing rates, and stimulation often resulted in further suppression of spiking.

Although stimulation artifacts resulted in amplitude threshold crossings that may be spuriously interpreted as a neuronal spike, we implemented several methods to mitigate the influence of nonphysiological activity. First, the characteristics of each unit (e.g. waveform shape) were manu-ally inspected during spike sorting and further quantified using several quality control metrics (e.g. interspike intervals); stimulation resulted in a stereotyped response that was easily detectable and removed from subsequent analyses. Additionally, we tested for modulation both during stimulation and in the post-stimulation epochs—a period in which no artifact was present. Contrary to what would be expected if stimulation artifacts were falsely elevating firing rates, we observed predominantly suppression during stimulation and enhancement post-stimulation.

Since we collected our microelectrode recordings in the context of a visual recognition memory task, we tested whether stimulation resulted in a change in memory performance. Although we hypothesized that stimulation would improve memory performance, we did not identify an apparent stimulation-related memory enhancement when controlling for individual differences, in contrast to our prior work (*Inman et al., 2018*). The absence of such an effect may be, at least in part, attributable to the considerable variability that we observed in this cohort; indeed, baseline memory performance and individual differences have been shown to account for a substantial portion of the variability in this amygdala-mediated memory enhancement effect (*Hollearn et al., 2025*).

Several studies on rats have demonstrated that brief electrical stimulation of the BLA can prioritize the consolidation of specific memories (*Bass et al., 2014*; *Bass et al., 2012*; *Bass and Manns, 2015*; *Manns and Bass, 2016*). These pro-memory effects emerged ~24 hr post-encoding and appear to be hippocampal-dependent (*Bass et al., 2014*), despite not resulting in a net change in the firing rates of hippocampal pyramidal neurons; instead, BLA stimulation resulted in brief periods of spike-field and field-field synchrony within CA3–CA1 in the low-gamma frequency range (30–55 Hz), which may facilitate spike-timing-dependent plasticity in recently active neurons (*Bass and Manns, 2015*).

The present study did not investigate interactions between spiking activity and local field poten-tials because neuronal spiking was sparse at baseline and often further suppressed by stimulation;

only a very small proportion of the total number of trials across all neurons exhibited ≥10 spikes in both the 1 s pre- and post-stimulation epochs (~2.5%). Although certain metrics are not biased by sample size (e.g. pairwise phase consistency), low spike counts can dramatically affect variance and, therefore, result in unstable estimates (*Vinck et al., 2011*).

How exactly the activity of single neurons is aggregated to produce local field potentials, which in turn interact with neuronal ensembles distributed throughout the brain, remains an active area of research (*Herreras, 2016*; *Kajikawa and Schroeder, 2011*; *Manning et al., 2009*; *Teleńczuk et al., 2020*). One recent study that leveraged closed-loop stimulation targeting memory consolidation during sleep observed neuronal spiking with greater phase-locking to medial temporal lobe slow-wave activity following stimulation (*Geva-Sagiv et al., 2023*); neuronal phase-locking, particularly to hippocampal theta oscillations, has long been associated with robust memory encoding and retrieval (*Jacobs et al., 2007*; *Rutishauser et al., 2010*; *Schonhaut et al., 2024*; *Yoo et al., 2021*). Further characterization of these spike-field interactions and refinement of closed-loop stimulation methods may provide a means for precisely modulating neuronal dynamics, for example, by entraining neuronal spiking that is phase-aligned to endogenous hippocampal theta oscillations to selectively enhance the encoding or retrieval of memories (*Hasselmo, 2005*; *Hasselmo et al., 2002*; *Hasselmo and Stern, 2014*; *Siegle and Wilson, 2014*).

Finally, we performed an exploratory analysis of neuronal pseudo-population activity, which suggests that hippocampal neurons exhibit robust changes in firing rate coactivity in response to BLA stimulation. Related research has similarly described how BLA stimulation can induce synchronous firing of hippocampal neurons, which has memory-enhancing effects (*Bass and Manns, 2015*). Other studies have leveraged similar, low-dimensional representations of population dynamics to describe how coordinated neural activity facilitates inferential reasoning and memory retrieval within the medial frontal cortex and hippocampus, respectively (*Courellis et al., 2024*; *Minxha et al., 2020*). Thus, a greater understanding of how neuronal coactivity may be precisely modulated by stimulation may help to refine therapeutic interventions targeting complex cognition and computation.

## Conclusions

By characterizing patterns of neuronal modulation evoked by intracranial TBS, we provide new insights that link micro- and macroscale responses to stimulation of the human brain. These insights advance our limited understanding of how focal electrical fields influence neuronal firing at the single-cell level and motivate future neuromodulatory therapies that aim to recapitulate specific patterns of activity implicated in cognition and memory.

## Methods

### Participants

We report results from a cohort of 23 patients with medically refractory epilepsy who underwent stereoelectroencephalography to localize epileptogenic foci (74% female, 19–66 years of age). All patients were age 18+ and able to provide informed consent. No exclusion was made concerning a patient's sex, gender, race, ethnicity, or socioeconomic status. Surgeries were performed at the University of Utah in Salt Lake City, UT, USA (n=10) and Barnes-Jewish Hospital in St. Louis, MO, USA (n=13). Patients were monitored continuously by a clinical team during their postoperative hospital course. Each patient signed a written informed consent form before participation in the research study; all study procedures were approved by the Institutional Review Board at the University of Utah (IRB 00144045, IRB 00114691) and Washington University (IRB 202104033).

### Electrode placement and localization

Numbers and trajectories of stereoelectroencephalography electrode placements were determined case-by-case and solely derived from clinical considerations during a multidisciplinary case conference without reference to this research program. Each patient was implanted with clinical macroelectrodes and 1–3 Behnke-Fried depth electrodes (Ad-Tech Medical Instrument Corporation, Oak Creek, WI, USA), which contained both macro- and microelectrode contacts (eight 40 µm diameter microwires and one unshielded reference wire) for recording local field potentials and extracellular action potentials, respectively (*Figure 1A*). To localize electrodes, we leveraged the open-access

*Localize Electrodes Graphical User Interface (LeGUI)* (*Davis et al., 2021*) software developed by our group, which performs coregistration of preoperative MRI and postoperative CT sequences, image processing, normalization to standard anatomical templates, and automated electrode detection.

## Intracranial electrophysiology

Neurophysiological data were recorded at both hospitals using a neural signal processor (Blackrock Microsystems, Salt Lake City, UT, USA; Nihon Koden USA, Irvine, CA, USA) sampling at 30 kHz. Microelectrode contacts were locally referenced to a low-impedance microwire near the recording wires. Macroelectrode contacts were referenced to an intracranial contact located within the white matter with minimal activity, per recommended practices (*Mercier et al., 2022*).

## Experimental design

Patients completed a visual recognition memory task previously employed by our group to characterize the effects of BLA stimulation upon memory consolidation (*Inman et al., 2018*). The memory task consisted of an encoding session, during which a series of neutral valence images were presented, and a self-paced retrieval session ~24 hr post-encoding wherein patients were asked to indicate whether each image onscreen was old (previously shown) or new (unseen) (*Figure 2A*). Data were aggregated across four highly similar experimental paradigms with minor differences in the content of visual stimuli, number of trials, stimulation parameters, and testing intervals. Each encoding session consisted of 160 or 320 trials wherein an image was presented on screen for 3 s, followed by an ~6 s interstimulus interval (fixation cross on screen).

## Spike detection and sorting

Microelectrode data were first filtered between 250 and 500 Hz with a zero-phase lag bandpass filter and re-thresholded offline at –3.5 times the root mean square of the signal to identify spike waveforms. Units were isolated during a semiautomated process within *Offline Sorter* (Plexon Inc, Dallas, TX, USA) by applying the T-distribution expectation maximization method on the first three principal components of the detected waveforms (initial parameters: degrees of freedom multiplier = 4, initial number of units = 3) (*Shoham et al., 2003*). Finally, the waveform shapes, interspike interval distribution, consistency of firing, and isolation from other waveform clusters were manually inspected for further curation and removal of spurious, nonphysiological threshold crossings that could represent stimulation artifacts.

## Single-unit quality metrics

We calculated several distinct metrics to characterize detected units' properties and assess the quality of our spike sorting (*Figure 1—figure supplement 2*): the number of units detected per microelectrode bundle, the mean firing rate (Hz) for each unit, the percentage of interspike intervals <3 ms, the coefficient of variation across each unit's spike train, the average presence ratio of firing in 1 s bins (proportion of bins which contain ≥1 spike), the ratio between the peak amplitude of the averaged waveform and its standard deviation, and the mean signal-to-noise ratio of the averaged waveform.

## Discrimination of excitatory vs. inhibitory neurons

We calculated two metrics from the averaged waveform from each detected unit: the VP and the PHW (*Figure 4A*); previously, these two properties of waveform morphology have been used to discriminate pyramidal cells (excitatory) from interneurons (inhibitory) in human intracranial recordings (*Peyrache et al., 2012*). Next, we performed k-means clustering (n=2 clusters) on the waveform metrics, in line with previous approaches to cell-type classification.

## Intracranial TBS

We delivered direct electrical stimulation to the BLA during half of the trials in the encoding phase of each experimental session. Stimulation pulses were delivered immediately once the image was removed from the screen and in a patterned rhythm designed to entrain endogenous theta-gamma oscillatory interactions (i.e. TBS) (*Hanslmayr et al., 2016*; *Suppa et al., 2016*). Specifically, we administered current-controlled, charge-balanced, bipolar, 1 mA, biphasic rectangular pulses over a 1 s period with a 50% duty cycle. Stimulation pulses were delivered at a rate of 50 Hz and nested within

eight equally spaced bursts (~8 Hz) (*Figure 2B and C*). A subset of experiments (n=7) used a lower current (0.5 mA) with variable pulse frequencies across trials (33 Hz, 50 Hz, 80 Hz).

## Peri-stimulation modulation analyses

We first created peri-stimulation epochs (1 s pre-trial ISI, 1 s after image onset, 1 s during stimulation/ after image offset, 1 s post-stimulation), with t=0 representing stimulation onset and the moment at which the image was removed from the screen (*Figure 2A*); identical epochs were created for the image-only (no-stimulation) trials. Each epoch was constrained to 1 s to ensure that subsequent firing rate contrasts were unbiased and to capture potential transient effects (e.g. image onset/offset). Units with a trial-averaged baseline (pre-trial ISI) firing rate of <0.1 Hz were excluded from subsequent analyses because low firing rates limited the ability to detect modulation robustly. Units were designated as 'modulated' if either the during- or post-stimulation firing rate contrast was significant following permutation testing (described in Statistical approach). An additional contrast of pre-trial ISI vs. image onset was performed to evaluate the sensitivity of neurons to task stimuli (i.e. image presentation).

To investigate whether stimulation altered rhythmicity in neuronal firing, we analyzed the spike timing autocorrelograms. More specifically, we computed the pairwise differences in spike timing for each trial (bin size = 5 ms, max lag = 250 ms) and then contrasted the differences in the latencies of the peak normalized autocorrelation value between epochs (pre-, during-, post-stimulation). Only neurons with a firing rate of ≥1 Hz (n=70/203, 34.5%) were included in this analysis since sparse firing resulted in noisy autocorrelation estimates.

Finally, we measured modulation in MUA by filtering the microelectrode signals in a 300–3000 Hz window and counting the number of threshold crossings. Thresholds were determined on a per-channel basis and defined as –3.5 times the root mean square of the signal during the baseline period; activity during stimulation was excluded since stimulation artifact is difficult to separate from MUA in the absence of spike sorting.

## Population analyses

To analyze pseudo-population activity, we performed a linear dimensionality reduction with principal component analysis (PCA) on a matrix of the z-scored trial-averaged firing rates of all neurons recorded across patients (*sklearn.decomposition.PCA*). This approach enables the identification of dominant patterns of coordinated neural activity that may not be apparent when examining individual neurons in isolation. Doing so allowed us to qualitatively examine the temporal dynamics of the dominant modes of neuron coactivity in a low-dimensional subspace, separated by experimental condition (stim vs. no-stim), region, and laterality relative to stimulation. The firing rate matrices for each condition were concatenated prior to PCA to facilitate direct comparison among the principal components; for simplicity, only the first three principal components are visualized. By collapsing across subjects into a common pseudo-population, this analysis provides a mesoscale view of how stimulation modulates shared activity patterns across anatomically distributed neural populations.

## Statistical approach

All statistical analyses were conducted using custom Python scripts and established statistical libraries (i.e. Scikit-learn [*Pedregosa et al., 2012*], *Scipy* [*Virtanen et al., 2020*], *Statsmodels* [*Seabold and Perktold, 2010*]). We performed two separate Wilcoxon signed-rank tests across trials on the during- and post-stimulation spike counts relative to their corresponding pre-trial baseline spike counts. To control for false positives, we compared the empirical test statistic against a null distribution generated from shuffling pre/during/post epoch labels (n=1000 permutations) (*Figure 2B*). An identical analysis was also performed on the no-stimulation (image-only) trials.

To test for differences in the proportion of modulated units (across conditions, regions, cell type, stimulation parameters, and behavioral outcomes), we performed a series of one- and two-sided Fisher's exact tests. Next, we used Mann-Whitney U tests to contrast baseline firing rates among modulated vs. unaffected units. Behavioral performance during the memory task was calculated using d-prime (d'), defined as the difference in an individual's z-scored hit rate and false alarm rate. Observed changes in recognition memory were split into two categories using a d' difference threshold of ±0.5: responder (Δd'<–0.5 or Δd' >+0.5) or non-responder (–0.5≤Δd'≤0.5). The threshold of ±0.5 was chosen based on the defined range of a 'medium effect' for Cohen's d, which bears conceptual similarity to

d'. To test the hypothesis that stimulation affected behavioral performance, we used a linear mixed effects model with d' score as the dependent variable, condition and experiment as fixed effects, and session as a random effect; an additional test for differences among hit rates (percent of previously seen images correctly identified) was implemented using a paired-samples t-test.

## Firing rate control analyses

We performed a series of control analyses to test whether our approach to firing rate detection was robust. First, we performed a sensitivity analysis by systematically varying the baseline firing rate threshold used to exclude units from modulation analyses. The threshold for inclusion of units was varied from 0 to 3 Hz (0.1 Hz step size), and the firing rate analyses were repeated to quantify the proportion of units meeting inclusion criteria and the proportion of units designated as modulated (*Figure 2—figure supplement 1A*). Next, we performed a dropout analysis wherein segments of data near the onset of a stimulation burst were removed from the during-stimulation epoch (an identical segment was also removed from the pre-trial ISI and post-stimulation epochs). To this end, we removed a window of data starting at the onset of each burst spanning 0–60 ms (5 ms step size, eight bursts in train) and recomputed the proportion of units meeting inclusion criteria and the proportion of units designated as modulated (*Figure 2—figure supplement 1B*). Finally, to better understand the trade-offs with our statistical approach, we generated synthetic data with different baseline firing rates (0.1–5.0 Hz) and effect sizes (±0.1–0.7 Hz), then simulated the likelihood of detecting modulation across variable conditions (*Figure 2—figure supplement 1C*).

## Acknowledgements

We are grateful to all the patients who participated in the study. This work was supported by the National Institute of Neurological Disorders and Stroke (T32NS115723; K23NS114178), the National Institute of Mental Health (R01MH120194), the National Science Foundation (NSF2124252, NSF1747505), and the Brain & Behavior Research Foundation (2023 NARSAD Young Investigator Grant). This study was conducted as part of a National Institutes of Health clinical trial (NCT05065450).

## Additional information

### Competing interests

Shuo Wang: Reviewing editor, *eLife*. The other authors declare that no competing interests exist.

### Funding

| Funder | Grant reference number | Author |
|---|---|---|
| National Institute of Neurological Disorders and Stroke | T32NS115723 | Justin M Campbell |
| National Institute of Neurological Disorders and Stroke | K23NS114178 | John D Rolston |
| National Institute of Mental Health and Neurosciences | R01MH120194 | Jon Timothy Willie |
| National Science Foundation | NSF2124252 | Cory Inman |
| National Science Foundation | NSF1747505 | Martina K Hollearn |
| Brain and Behavior Research Foundation | 2023 NARSAD Young Investigator Grant | Tao Xie |

The funders had no role in study design, data collection and interpretation, or the decision to submit the work for publication.

## Author contributions

Justin M Campbell, Conceptualization, Data curation, Software, Formal analysis, Investigation, Visualization, Methodology, Writing – original draft, Writing – review and editing; Rhiannon L Cowan, Conceptualization, Data curation, Writing – review and editing; Krista L Wahlstrom, Investigation, Project administration, Writing – review and editing; Martina K Hollearn, Dylan Jensen, Stephan Hamann, Writing – review and editing; Tyler Davis, Software, Writing – review and editing; Shervin Rahimpour, Ben Shofty, Amir Arain, John D Rolston, Shuo Wang, Lawrence N Eisenman, Resources, Writing – review and editing; James Swift, Software, Investigation, Writing – review and editing; Tao Xie, Investigation, Writing – review and editing; Peter Brunner, Software, Supervision, Investigation; Joseph Manns, Conceptualization, Supervision, Project administration, Writing – review and editing; Cory Inman, Resources, Supervision, Funding acquisition, Project administration, Writing – review and editing; Elliot H Smith, Conceptualization, Resources, Supervision, Writing – review and editing; Jon Timothy Willie, Resources, Supervision, Funding acquisition, Investigation, Project administration, Writing – review and editing

## Author ORCIDs

Justin M Campbell ⓘ https://orcid.org/0000-0002-8685-2081
Rhiannon L Cowan ⓘ https://orcid.org/0000-0002-8397-960X
Shuo Wang ⓘ https://orcid.org/0000-0003-2562-0225
Peter Brunner ⓘ https://orcid.org/0000-0002-2588-2754
Elliot H Smith ⓘ https://orcid.org/0000-0003-4323-4643

## Ethics

Clinical trial registration NCT05065450.
All study procedures were approved by the Institutional Review Board at the University of Utah (IRB 00144045, IRB 00114691) and Washington University (IRB 202104033).

Reviewer #1 (Public review): https://doi.org/10.7554/eLife.106481.3.sa1
Reviewer #2 (Public review): https://doi.org/10.7554/eLife.106481.3.sa2
Author response https://doi.org/10.7554/eLife.106481.3.sa3

# Additional files

## Supplementary files

MDAR checklist

Supplementary file 1. Patient demographics and clinical characteristics.

## Data availability

Custom Python analysis scripts used in the manuscript are publicly available on GitHub (https://github.com/Justin-Campbell/BLAESUnits copy archived at *Campbell, 2025*). The dataset analyzed in the study is available in an Open Science Foundation (OSF) repository under a Creative Commons Attribution 4.0 International license and can be accessed at: https://osf.io/y5j28/.

The following dataset was generated:

| Author(s) | Year | Dataset title | Dataset URL | Database and Identifier |
|---|---|---|---|---|
| Campbell J | 2025 | Data for "Human single-neuron activity is modulated by intracranial theta burst stimulation of the basolateral amygdala" | https://osf.io/y5j28/ | Open Science Framework, y5j28 |

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
