## [Editor Report · eLife Assessment]

This **important** study provides a description of how single-neuron firing rates in the human medial temporal lobe and frontal cortex are modulated by theta-burst stimulation of the basolateral amydala. The results are supported by **convincing** evidence obtained from a rigorous task design and analysis of an incredibly rare dataset. The results may help guide future studies incorporating amygdala stimulation to improve patient health.

---

## [Referee Report · Reviewer #1 (Public review)]

In this manuscript, Campbell et al. assess how intracranial theta-burst stimulation (TBS) applied to the basolateral amygdala in 23 epilepsy patients affects neuronal spiking in the medial temporal lobe and prefrontal cortex during a visual recognition memory task. This is an incredibly rare dataset; collecting single-unit spiking data from behaving humans during active intracranial stimulation is a Herculean task, with immense potential for translational studies of how stimulation may be applied to modulate biological mechanisms of memory. The authors utilize careful, high quality methodology throughout (e.g. task design, spike recording and sorting, statistical analysis), providing high confidence in the validity of their findings.

In providing such a detailed and deep investigation into the single-unit responses to intracranial stimulation the authors provide a very useful resources to any researchers in the fields of brain stimulation and human neurophysiology. This work could be instrumental in guiding diverse research studies, from basic science investigating the role of theta oscillations in human cognition to translational work investigating deep-brain stimulation for memory.

The authors have adequately addressed all prior concerns.

---

## [Referee Report · Reviewer #2 (Public review)]

Summary:

This study presents a valuable characterization of the effects of intracranial theta-burst stimulation of the basolateral amygdala on single units spiking activity in several areas in the human brain, associated with memory processing. It is written clearly and concisely, allowing readers to fully understand the analysis used.

The authors used a visual recognition memory task previously employed by their group to characterize the effects of basolateral amygdala stimulation upon memory consolidation (Inman et al, 2018). This current report presents an interesting analysis that complements the results reported in the 2018 paper.

Strengths:

Rare combination of human neurophysiology and behavior -

The type of experiment performed in the manuscript, which contains both neurophysiological data, behavior, and a deep brain stimulation intervention (DBS), is incredibly rare, takes many years to accomplish with tight collaboration between clinical and research teams. Our understanding of spiking dynamics of human neurons is very limited, and this report is an important piece in the puzzle that allows DBS to be used in future interventions that will benefit patients' health.

Multiple brain areas included -

It's important to note that the report analyzes brain areas with which the Amygdala has extensive connections (Fig. 1A) - Hippocampus, OFC, Amygdala, ACC. It seems that neurons in all these areas were modulated by the stimulation, except the ACC, in which firing rates were so low that only a handful of neurons were included in the analysis. This is an important demonstration that low-amplitude stimulation (even when reduced to 0.5mA) can travel far and wide across the human brain.

The experiment is cleverly designed to tease apart responses due to visual stimuli (image presentation) and electrical stimulation. Authors suggest that the units modulated by stimulation are largely distinct from those responsive to image offset during trials without stimulation. The subpopulation that responds strongly also tends to have a higher baseline firing rate. It's important to add that the chosen modulation index is more likely to be significant in neurons with higher firing rates (Figure S8). The authors discuss the tradeoff of using a nonparametric modulation index for vs. other methods (for example, percent change in trial-averaged firing rate from baseline).

---

## [Author Response]

The following is the authors’ response to the original reviews.

**Reviewer #1 (Public review):**
**“**This is an exploratory study that doesn't explore quite enough. Critically, the authors make a point of mentioning that neuronal firing properties vary across cell types, but only use baseline firing rate as a proxy metric for cell type. This leaves several important explorations on the table, not limited to the following:”1a: “Do waveform shape features, which can also be informative of cell type, predict the effect of stimulation?”

To address this question, we modeled our approach to cell type classification after Peyrache et al. 2012. More specifically, we extracted two features from the mean unit waveforms—the valley-to-peak time (VP) and the peak half-width (PHW). These features were then used to classify units into two distinct clusters (k-means, clusters = 2, based on a strong prior from existing literature), representing putative excitatory and inhibitory neurons. Our approach recapitulated many of the same observations in Peyrache et al. 2012, namely (1) identification of two clusters (low PHW/VP: inhibitory, high PHW/VP: excitatory), (2) an ~80/20 ratio of excitatory/inhibitory neurons, and (3) greater baseline firing rates in the inhibitory vs. excitatory neurons. However, we did not observe a preferential modulation of one cell type compared to another (see newly created Figure 4). A description of this analysis and its takeaways has been incorporated into the manuscript.

Change to Text:

Created Figure 4 (Separation of presumed excitatory and inhibitory neurons by waveform morphology).

Caption: (A) Two metrics were calculated using the averaged waveforms for each detected unit: the valley-to-peak width (VP) and peak half-width (PHW). (B) Scatterplot of the relationship between VP and PHW; note that units with identical metrics are overlaid. Using k-means clustering, we identified two distinct response clusters, representing presumed excitatory (E, blue) and inhibitory (I, red) neurons. The units from which the example waveforms were taken are outlined in black. Probability distributions for each metric are shown along the axes. (C) Total number of units within each cluster, separated by region. (D) Comparison of baseline firing rates, separated by cluster. (E) Percent of modulated units in each cluster. * p < 0.05, NS = not significant.

Added a description of clustering methodology to lines 132-137: “We calculated two metrics from the averaged waveform from each detected unit: the valley-to-peak-width (VP) and the peak half-width (PHW) (Figure 4A); previously, these two properties of waveform morphology have been used to discriminate pyramidal cells (excitatory) from interneurons (inhibitory) in human intracranial recordings (Peyrache et al., 2012). Next, we performed k-means clustering (n = 2 clusters) on the waveform metrics, in line with previous approaches to cell type classification.

Added a section in the Results titled “Theta Burst Stimulation Modulates Excitatory and Inhibitory Neurons Equally”. Lines 370-378: “Using k-means clustering, we grouped neurons into two distinct clusters based on waveform morphology, representing neurons that were presumed to be excitatory (E) and inhibitory (I) (Figure 4B). Inhibitory (fast-spiking) neurons exhibited shorter waveform VP and PHW, compared with excitatory (regular-spiking) neurons (I cluster centroid: VP = 0.50ms, PHW = 0.51ms; E cluster centroid: VP = 0.32ms, PHW = 0.31ms), and greater baseline firing rates (U(N_I_ = 23, N<_E_ = 133) = 1074.50, p = 0.023) (Figure 4D). Although we observed a much greater proportion of excitatory vs. inhibitory neurons (E: 85.3%, I: 14.7%), stimulation appeared to affect excitatory and inhibitory neurons equally, suggesting that one cell type is not preferentially activated over another (Figure 4E).

Modified discussion of the effects of stimulation on different cell types. Lines 475-483: “…To test these hypotheses directly, we clustered neurons into presumed excitatory and inhibitory neurons based on waveform morphology. In doing so, we observed ~85% excitatory and ~15% inhibitory neurons, which is very similar what has been reported previously in human intracranial recordings (Cowan et al. 2024, Peyrache et al., 2012). Interestingly, stimulation appeared to modulate approximately the same proportion of neurons for each cell type (~30%), despite the differently-sized groups. Recent reports, however, have suggested that the extent to which electrical fields entrain neuronal spiking, particularly with respect to phase-locking, may be specific to distinct classes of cells (Lee et al., 2024).”

1b: “Is the autocorrelation of spike timing, which can be informative about temporal dynamics, altered by stimulation? This is especially interesting if theta-burst stimulation either entrains theta-rhythmic spiking or is more modulatory of endogenously theta-modulated units.”

The reviewer is correct in suggesting that rate-modulation represents only one of many possible ways by which exogenous theta burst stimulation may influence neuronal activity. Indeed, intracranial theta burst stimulation has previously been shown to evoke theta-frequency oscillatory responses in local field potentials (Solomon et al. 2021), and other forms of stimulation (i.e., transcranial alternating current stimulation) may modulate the rhythm, rather than the rate, of neuronal spiking (Krause et al. 2019).

To investigate whether stimulation altered rhythmicity in neuronal firing, we contrasted the spike timing autocorrelograms, as suggested. More specifically, we computed the pairwise differences in spike timing for each trial, separating spikes into the same pre-, during-, and post-stimulation epochs described in the manuscript (bin size = 5 ms, max lag = 250 ms), grouped neurons by whether they were modulated, and then contrasted the differences in the latencies of the peak normalized autocorrelation value between epochs. Only neurons with a firing rate of ≥ 1 Hz (n = 70/203, 34.5%) were included in this analysis since sparse firing resulted in noisy autocorrelation estimates. Subsequent statistical testing of the peak latency differences between pre-/during- and pre-/post-stimulation did not reveal any group-level differences (Mann-Whitney U tests, p > 0.05). Thus, we were not able to identify neuronal responses suggestive of altered rhythmicity (see Figure S5). A description of this analysis and its takeaways has been incorporated into the manuscript.

Of note, there are two elements of the data that constrain our ability to detect modulation in the rhythm of firing. First, the baseline activity recorded across neurons modulated by stimulation was relatively low (i.e., median firing rate = 1.77 Hz). Second, stimulation often resulted in a suppression, rather than an enhancement, of firing rate. Taken together, the sparse firing afforded limited opportunity to characterize changes to subtle patterns of spiking.

Change to Text:

Created Figure S5 (Analysis of modulation in spiking rhythmicity)

Caption: (A) Representative autocorrelograms ACG for a single neuron. The pairwise differences in spike timing were computed for each trial and epoch (bin size = 5 ms, max lag = 250 ms), then smoothed with a Gaussian kernel. The peak in the normalized ACG across trials was computed for each epoch. (B) Kernel density estimate of the peak ACG lag, separated by epoch. (C) The peak ACG lags were split by whether the neuron was modulated (Mod) or unaffected by stimulation (NS = not significant) for each of the two contrasts: pre- vs. during-stim (left) and pre- vs. post-stim (right).

Details about the autocorrelation methodology have been incorporated. Lines 166-172: “To investigate whether stimulation altered rhythmicity in neuronal firing, we analyzed the spike timing autocorrelograms. More specifically, we computed the pairwise differences in spike timing for each trial (bin size = 5 ms, max lag = 250 ms) and then contrasted the differences in the latencies of the peak normalized autocorrelation value between epochs (pre-, during-, post-stimulation). Only neurons with a firing rate of ≥ 1 Hz (n = 70/203, 34.5%) were included in this analysis since sparse firing resulted in noisy autocorrelation estimates.

The results from contrasting the autocorrelograms are now mentioned briefly. Lines 297-298: “Stimulation, however, did not appear to alter the rhythmicity in neuronal firing, as measured by spiking autocorrelograms (Figure S5).”

1c: “The authors reference the relevance of spike-field synchrony (30-55 Hz) in animal work, but ignore it here. Does spike-field synchrony (comparing the image presentation to post-stimulation) change in this frequency range? This does not seem beyond the scope of investigation here.”

We agree that a further characterization of spike-field and spike-phase relationships may provide rich insights into more complex regional and interregional dynamics that may be altered by stimulation. Given that many metrics are biased by sample size (e.g., number of spikes), which can vary considerably, computing the pairwise phase consistency (PPC) between spikes and LFP is a preferred metric (Vinck et al. 2010). Although PPC is unbiased, its variance nonetheless increases considerably with low spike counts; pooling spike counts across trials, however, decouples the temporal relationship between spiking and the LFP phase for each trial, confounding results and yielding an unstable estimate.

To determine whether such an analysis is indeed possible, we calculated the percentage of stimulation trials with ≥ 10 spikes in both the 1s pre- and post-stimulation epochs (a relatively low threshold for inclusion). Only a very small proportion of the total number of trials across all neurons met this criterion (2.5%). Thus, because of the sparse spiking in our data, we are unable to reliably characterize spike-field or spike-phase modulation in detected neurons.

Change to Text:

In the manuscript, we have added a description of why our data is not well-suited to investigate these relationships.

Lines 532-538: “The present study did not investigate interactions between spiking activity and local field potentials because neuronal spiking was sparse at baseline and often further suppressed by stimulation; only a very small proportion of the total number of trials across all neurons exhibited ≥ 10 spikes in both the 1s pre- and post-stimulation epochs (~2.5%). Although certain metrics are not biased by sample size (e.g., pairwise phase consistency), low spike counts can dramatically affect variance and, therefore, result in unstable estimates (Vinck et al., 2011).

1d: “How does multi-unit activity respond to stimulation? At this somewhat low count of neurons (total n=156 included) it would be valuable to provide input on multi-unit responses to stimulation as well.”

We thank the reviewer for this suggestion. We have incorporated an analysis of multiunit activity (MUA), which similarly identifies robust modulation via permutation-based statistical testing and characterizes the different profiles of responses (i.e., increased vs. decreased MUA threshold crossings pre- vs. post-stimulation).

Change to Text:

Created Figure S8 (Analysis of multiunit activity response to stimulation)

Caption: (A) Example trace of multiunit activity (MUA) in one channel during a single stimulation trial. Threshold crossings are highlighted with a pink dot overlaid on the MUA signal with a corresponding hash below. (B) The percentage of channels with significantly modulated MUA, separated by the direction of effect. (C) The percentage of channels with significantly modulated MUA, separated by direction effect and region. Inc (red; post > pre) vs. Dec (blue; post < pre). HIP = hippocampus, OFC = orbitofrontal cortex, AMY = amygdala, ACC = anterior cingulate cortex. *** p < 0.001, NS = not significant.

Details about the MUA methodology have been incorporated. Lines 174-180: “Finally, we measured modulation in multiunit activity (MUA) by filtering the microleectrode signals in a 300-3,000 Hz window and counting the number of threshold crossings. Thresholds were determined on a per-channel basis and defined as -3.5 times the root mean square of the signal during the baseline period; activity during stimulation was excluded since stimulation artifact is difficult to separate from MUA in the absence of spike sorting.

MUA results are now incorporated. Lines 365-367: “Additional characterization of MUA revealed a dominant signature of increased activity post- vs. pre-stimulation, in line with these trends observed at the single-neuron level (Figure S8).”

1e: “Several intracranial studies have implicated proximity to white matter in determining the effects of stimulation on LFPs; do the authors see an effect of white matter proximity here?”

We thank the reviewer for the interesting question. Subsequent characterization revealed only small differences in the proximity of stimulation contacts to white matter (range 1.5-8.0 mm), likely because the chosen target (i.e., basolateral amygdala) has several nearby white matter structures (e.g., stria terminalis). Nonetheless, we performed a linear regression between the proximity to white matter and the stimulation-induced effect on behavior (stimulation vs. no-stimulation d’ difference), the results of which indicate no clear association (p > 0.05; see Figure S9). Critically, this is not to suggest that white matter proximity has no interaction with the reported behavioral effects, but rather, that we could not identify such an association within our data.

Change to Text:

Created Figure S9 (The effect of stimulation proximity to white matter and distance to recorded neurons).

Caption: (A) Kernel density estimate of the Euclidean distance from stimulation contacts to nearest WM structure (in mm); hash marks represent individual observations. (B) The change in memory performance (Δd’) was linearly regressed onto the distance from the stimulated contacts to white matter.

The following has been added to lines 405-426: “Proximity to white matter has been shown to influence the effects of stimulation on behavior and the strength of evoked responses (Mankin et al., 2021; Mohan et al., 2020; Paulk et al., 2022). Across all stimulated contacts, we observed only small differences in the proximity of stimulation contacts to white matter (median = 4.5 mm, range = 1.5-8.0 mm), likely because the chosen target (i.e., basolateral amygdala) has several nearby white matter structures (e.g., stria terminalis). Nonetheless, we performed a linear regression between the proximity to white matter and the stimulation-induced effect on behavior (stimulation vs. no-stimulation d’ difference), the results of which indicate no clear association (p > 0.05; see Figure S9).

Comment 2: “It is a little confusing to interpret stimulation-induced modulation of neuronal spiking in the absence of stimulation-induced change in behavior. How do the authors findings tell us anything about the neural mechanisms of stimulation-modulated memory if memory isn't altered? In line with point #1, I would suggest a deeper dive into behavior (e.g. reaction time? Or focus on individual sessions that do change in Figure 4A?) to make a stronger statement connecting the neural results to behavioral relevance.”

We agree that the connection between the observed stimulation-induced neuronal modulation and effects on behavior is unclear and has proven challenging to elucidate. Per the reviewer’s suggestion, we further focused our analyses on the neuronal modulation effects in the individual sessions that resulted in a robust change in memory performance (stimulation vs. no-stimulation d’ difference threshold of ± 0.5, based on a moderate effect size for Cohen’s d); both a positive and negative threshold were used to capture robust changes in memory performance associated with firing rate modulation, whether enhancement or suppression. To this end, we contrasted the proportion of modulated neurons in the sessions where stimulation resulted in a robust behavioral change (Δd’) with those that did not (~d’). We did not observe a difference in the proportions between groups when collapsed across all sampled regions, or when separately evaluated (Fisher’s exact tests, p > 0.05; see Figure 5C).

Given that this approach did not further clarify the connection between our neural and behavioral results, we believe it is most appropriate to deemphasize claims in the manuscript regarding the potential insights for behavioral modulation (e.g., memory enhancement), and have done so.

Change to Text:

Toned down reference to the memory-related effects of stimulation in the abstract by removing the following lines from the abstract: “Previously, we demonstrated that intracranial theta burst stimulation (TBS) of the basolateral amygdala (BLA) can enhance declarative memory, likely by modulating hippocampal-dependent memory consolidation…” and “…and motivate future neuromodulatory therapies that aim to recapitulate specific patterns of activity implicated in cognition and memory.”

Changed Figure 4 to Figure 5

Created Figure 5C (Interaction between behavioral effects and neuronal modulation)(C) Change in recognition memory performance was split into two categories using a d’ difference threshold of ± 0.5: responder (positive or negative; Δd’, pink) and non-responder (~d’, grey). Individual d’ scores are shown (left) with points colored by outcome category; dotted lines demarcate category boundaries, and the grey-shaded region represents negligible change. The number of sessions within each outcome category (middle) and the proportion of modulated units as a function of outcome category, separated by region (right). NS = not significant.

The description of the behavioral results has been updated. Lines 394-403: “At the level of individual sessions, we observed enhanced memory (Δd’ > +0.5) in 36.7%, impaired memory (Δd’ < -0.5) in 20.0%, and negligible change (-0.5 ≤ Δd’ ≤ 0.5) in 43.3% when comparing performance between the stim and no-stim conditions; a threshold of Δd’ ± 0.5 was chosen for this classification based on the defined range of a “medium effect” for Cohen’s d. To test our hypothesis that neuronal modulation would be associated with changes in memory performance, we combined the sessions that resulted in either memory enhancement or impairment and contrasted the proportion of modulated units across regions sampled. We did not, however, observe a meaningful difference in the proportion of modulated units when grouped by behavioral outcome (all contrasts p > 0.05) (Figure 5C).

Lines 213-214 and 394-397 have been edited to reflect a change in the d’ threshold used for categorizing behavioral results (from Δd’ ± 0.2 to Δd’ ± 0.5).

Comment 3: “It is not clear to me why the assessment of firing rates after image onset and after stim offset is limited to one second - this choice should be more theoretically justified, particularly for regions that spike as sparsely as these.”

We thank the reviewer for this question and acknowledge that no clear justification was provided for this decision in the manuscript. Our decision to limit each of the analysis epochs to 1s was chosen for two reasons. First, the maximum possible length of the during-stimulation epoch was 1 s (stim on for 1 s). Although the pre- and post-stimulation epochs could be extended without issue, we were concerned that variable time windows could introduce a bias, for instance, resulting in different variances between epochs. Second, we anticipated, both from empirical observations and prior literature, that the neural response following stimulation or task features (e.g., image onset/offset) was likely to be transient, rather than sustained for a period of many seconds. By keeping the windows short, we ensured that our approach to detecting modulation (i.e., contrasting trial-wise spike counts between each pair of epochs) captured the intended effect rather than random noise. We have incorporated a discussion of this rationale in the Peri-Stimulation Modulation Analyses section.

Change to Text:

Lines 156-158 have been added: “Each epoch was constrained to 1 s to ensure that subsequent firing rate contrasts were unbiased and to capture potential transient effects (e.g., image onset/offset).”

Comment 4: “This work coincides with another example of human intracranial stimulation investigating the effect on firing rates (doi: https://doi.org/10.1101/2024.11.28.625915). Given how incredibly rare this type of work is, I think the authors should discuss how their work converges with this work (or doesn't).”

Thank you for bringing this highly relevant work to our attention. We were unaware of this recent preprint and have incorporated a discussion of its main findings into the manuscript.

Change to Text:

New citations: van der Plas et al. 2024 (bioRxiv), Cowan et al. 2024 (bioRxiv)

The discussion of related studies has been updated. Lines 447-457: “Few studies, however, have characterized the impact of electrical stimulation via macroelectrodes on the spiking activity of human cortical neurons, none of which involve intracranial theta burst stimulation. One study reported a long-lasting reduction in neural excitability among parietal neurons, with variable onset time and recovery following continuous transcranial TBS in non-human primates (Romero et al., 2022). In a similar vein, it was recently shown that human neurons are largely suppressed by single-pulse electrical stimulation (Cowan et al., 2024; Plas et al., 2024). Other emerging evidence suggests that transcranial direct current stimulation may entrain the rhythm rather than rate of neuronal spiking (Krause et al., 2019) and that stimulation-evoked modulation of spiking may meaningfully impact behavioral performance on cognitive tasks (Fehring et al., 2024).”

Comment 5: “What information does the pseudo-population analysis add? It's not totally clear to me.”

We recognize the need to further contextualize the motivation for the exploratory pseudo-population analysis and appreciate the reviewer for bringing the lack of detail to our attention. In brief, the analysis allowed us to observe trends in activity across populations of neurons, which, in principle, are not visible by characterizing modulation solely in discrete neurons. Additional details have been incorporated into the manuscript, as suggested.

Change to Text:

Additional justification has been incorporated in the description of the methodology. Lines 185-187: “…This approach enables the identification of dominant patterns of coordinated neural activity that may not be apparent when examining individual neurons in isolation.”, lines 192-194: “…By collapsing across subjects into a common pseudo-population, this analysis provides a mesoscale view of how stimulation modulates shared activity patterns across anatomically distributed neural populations.”

A summary interpretation has been added to the paragraph describing the results. Lines 326-328: “Taken together, these analyses reveal global structure in the state space of responses to BLA stimulation within hippocampal circuits.”

**Reviewer #2 (Public review):**
Comment 1 “Authors suggest that the units modulated by stimulation are largely distinct from those responsive to image offset during trials without stimulation. The subpopulation that responds strongly also tends to have a higher baseline of firing rate. It's important to add that the chosen modulation index is more likely to be significant in neurons with higher firing rates.”

This is an important point that was not previously addressed in our manuscript. We suspect there are likely two factors at play worth considering with respect to our chosen nonparametric modulation index: neurons with lower activity require smaller changes in spike counts to be significantly modulated (easier to flip ranks), and neurons with higher activity empirically exhibit greater absolute shifts in the number of spikes. Our further use of permutation testing, while mitigating false positives, may also somewhat constrain the ability to detect modulation in sparsely active neurons. Nonetheless, given that many trials entailed few or no spikes, we believe this approach is preferable to alternatives that may be more susceptible to noise (e.g., percent change in trial-averaged firing rate from baseline).

To better understand the tradeoffs with detection probability, we performed a sensitivity analysis. We generated synthetic data with different baseline firing rates (0.1-5.0 Hz) and effect sizes (± 0.1-0.7 Hz) and simulated the likelihood of detection with our given modulation index across neurons. The results of the simulation support the notion that the probability of detecting modulation is lower for sparsely active neurons (Figure S8C). Further discussion of this consideration for the chosen modulation index, as well as details regarding the sensitivity analysis, have been incorporated into the manuscript.

Change to Text:

Created Figure S7C (Detection probability analysis)

Caption: The same permutation-based analyses reported in the manuscript were repeated under different control conditions… (C) Visualization of the predicted probability of detecting modulation across synthetic neurons with variable firing rates and modulation effect sizes; FR = firing rate.

Lines 223-224 have been added to the Methods section titled “Firing Rate Control Analyses”: “We performed a series of control analyses to test whether our approach to firing rate detection was robust…”

A description of the simulation has been incorporated into the same section as above. Lines 234-237: “Finally, to better understand the tradeoffs with our statistical approach, we generated synthetic data with different baseline firing rates (0.1-5.0 Hz) and effect sizes (± 0.1-0.7 Hz), then simulated the likelihood of detecting modulation across variable conditions (Figure S7C).”

The description of the results from the control analyses has been updated. Lines 330-339: “Finally, we performed three supplementary analyses to evaluate the robustness of our approach to detecting firing rate modulation: a sensitivity analysis assessing the proportion of modulated units at different firing rate thresholds for inclusion/exclusion, a data dropout analysis designed to control for the possibility that non-physiological stimulation artifacts may preclude the detection of temporally adjacent spiking, and a synthetic detection probability analysis. These results recapitulate our observation that units with higher baseline firing are most likely to exhibit modulation (though the probability of detecting modulation is lower for sparsely active neurons) and suggest that suppression in firing rate is not solely attributable to amplifier saturation following stimulation (Figure S7).

Comment 2: “Readers can benefit from understanding with more details the locations chosen for stimulation - in light of previous studies that found differences between effects based on proximity to white matter (For example - PMID 32446925, Mohan et al, Brain Stimul. 2020 and PMID 33279717 Mankin et al Brain Stimul. 2021).”

This has been addressed in the above response to Reviewer’s 1 comment 1.1e.

Change to Text:

See changes related to Reviewer 1 comment 1.1e.

Comment 3: “Missing information in the manuscript…”3a: “Images of stimulation anatomical locations for all subjects included in this study. Ideally information about the impedance of the contacts to be able to calculate the actual current used.”

As requested, we have provided an image from the coronal T1 MRI sequence, which highlights the position of the stimulated contacts for each of the 16 patients. Though we did not measure the impedances directly, the stimulation was current-controlled, which ensured that the desired current and charge density were consistent regardless of the tissue or electrode impedance.

Change to Text:

Created Figure S1 (Anatomical location of stimulated electrodes).

Caption: A coronal slice from the T1-weighted MRI scan is shown for each patient who participated in the study (n = 16). Electrode contacts within the same plane of the image are shown with blue circles, and the bipolar pair of stimulated contacts within the basolateral amygdala is highlighted in red.

Lines 144-145 have been edited to reflect that the delivered stimulation was current-controlled: “Specifically, we administered current-controlled, charge-balanced, …”

3b: “The studied population is epilepsy patients, and the manuscript lacks description of their condition, proximity to electrodes included in the study to pathological areas, and the number of units from each patient/hemisphere.”

We agree that additional information regarding patient demographics, experimental details, and clinical characteristics would further contextualize this unique patient population. A new table has been included, which contains the following information: patient ID, sex, age, # experimental session, # SEEG leads (and # microelectrodes), # detected units (L vs. R hemisphere), and suspected seizure onset zone.

Change to Text:

Created Table S1 (Patient demographics and clinical characteristics).

Lines 258-259 have been added: “…(see Table S1 for patient demographics).”

3c: “I haven't seen any comments on code availability (calculating modulation indices and statistics) and data sharing.”

For clarification, a section titled Resource Availability is already appended to the end of the manuscript following the Conclusion, which describes the data and code availability.

Change to Text:

None

3d: “Small comment - Figure legend 3E - Define gray markers (non-modulated units?)”

Thank you for highlighting this omission. We have updated the relevant figure caption.

Change to Text:

The following has been added to the Figure 3 caption: “…whereas units without a significant change in activity are shown in grey.”